# The Prevalence and Comorbidity of Tic Disorders and Obsessive-Compulsive Disorder in Chinese School Students Aged 6–16: A National Survey

**DOI:** 10.3390/brainsci12050650

**Published:** 2022-05-16

**Authors:** Junjuan Yan, Hu Deng, Yongming Wang, Xiaolin Wang, Tengteng Fan, Shijie Li, Fang Wen, Liping Yu, Fang Wang, Jingran Liu, Yuanzhen Wu, Yi Zheng, Yonghua Cui, Ying Li

**Affiliations:** 1Department of Psychiatry, Beijing Children’s Hospital, Capital Medical University, National Centre for Children’s Health, Beijing 100045, China; yanzhen0328@163.com (J.Y.); wenfang0812@163.com (F.W.); dongyuzui1@163.com (L.Y.); doramon0807@sina.cn (F.W.); liujingran123@sina.com (J.L.); 2Department of Innovation and Transformation, Peking University HuiLongGuan Clinical Medical School, Beijing Huilongguan Hospital, Beijing 100096, China; denghu501@163.com; 3School of Biology & Basic Medical Sciences, Medical College of Soochow University, Suzhou 215123, China; wangyongming16@mails.ucas.ac.cn; 4Laboratory of Tumor Immunology, Beijing Pediatric Research Institute, Beijing Children’s Hospital, Capital Medical University, National Center for Children’s Health, Beijing 100045, China; wangxl19891012@163.com; 5Peking University Sixth Hospital, Peking University Institute of Mental Health, NHC Key Laboratory of Mental Health (Peking University), National Clinical Research Center for Mental Disorders (Peking University Sixth Hospital), Beijing 100191, China; fantengteng@bjmu.edu.cn; 6Department of Child Health Care, Beijing Children’s Hospital, Capital Medical University, National Center for Children’s Health, Beijing 100045, China; shijie_liccmu@163.com; 7Beijing Anding Hospital, Capital Medical University, Ankang Hutong, Beijing 100101, China; wyz13718397073@126.com (Y.W.); yizheng@ccmu.edu.cn (Y.Z.)

**Keywords:** China, tic disorders, obsessive-compulsive disorder, prevalence, children, adolescents

## Abstract

Background: Obsessive-compulsive disorder (OCD) and tic disorders (TDs) are closely related and considered to etiologically overlap. Both disorders are characterized by repetitive behaviors. TD and OCD often co-occur. The high comorbidity between OCD and TD individuals suggests that we also need to pay more attention to the homogeneity and heterogeneity between TS and OCD. To date, there has been no systematic nationwide epidemiological survey of the mental health (including tic disorders and obsessive-compulsive disorder) of children and adolescents in China. Methods: A two-stage epidemiological study of psychiatric point prevalence was conducted. We used the multistage cluster stratified random sampling strategy to assess five provinces of China. The Child Behavior Checklist was used to identify behavioral problems among the enrolled students in the first stage. The results from the Mini-International Neuropsychiatric Interview for Children and Adolescents and evaluations from two psychiatrists based on the Diagnostic and Statistical Manual-IV were used to make a diagnosis. Point weighted prevalence for TD and OCD was estimated. We adjusted prevalence estimates with the product of sampling weights and poststratification weights. Standard error values and 95% confidential intervals were generated with Taylor series linearization. Rao–Scott adjusted chi-square (*χ*^2^) tests were employed to compare the prevalence estimates of different age and sex groups. Results: In the first stage, 73,992 participants aged 6–16 years old were selected. The prevalence rates of OCD and TDs were 1.37% (95% CI: 1.28–1.45) and 2.46% (95% CI: 2.35–2.57), respectively. The prevalence of OCD was found to be higher in girls (*p* < 0.001) and higher in boys with transient tic disorder (TTD) (*p* < 0.001) and Tourette’s syndrome (TS) (*p* < 0.001). The most common comorbidity of TS was OCD (40.73%), and for OCD, it was TS (11.36%). Conclusions: Our study is the first nationwide survey on the prevalence of TD (2.46%) and OCD (1.37%) in school students aged 6–16 years old in China. The high comorbidity between OCD and TD individuals suggested overlap based on the prevalence dimensions, which might be influenced by age and sex. This result suggested that we also need to pay more attention to the homogeneity and heterogeneity between TS and OCD.

## 1. Introduction

Tic disorders (TDs) are common neurodevelopmental disorders in children and adolescents. Tourette’s syndrome (TS), chronic motor tic disorder (CMTD), chronic vocal tic disorder (CVTD), and transient tic disorder (TTD) are the main diagnostic types of TDs [1]. Obsessive-compulsive disorder (OCD) is a neurotic disorder characterized by the presence of obsessions and/or compulsions [1]. TD and OCD often co-occur and overlap to some extent. Some difficulties exist in symptom recognition, especially when borderline symptoms, such as repetitive movements, are present [2]. OCD is highly prevalent among patients with TS, with lifetime rates of 50.0% [3]. “Tic-related OCD” represents a phenomenologically characteristic subtype of OCD [1]. Although TD often improves in adulthood, comorbid OCD may tend to persist and may cause significant adverse effects on quality of life [4,5]. This indicates that the comorbid condition of TD may affect the prognosis more than the disease itself.

Children and adolescents with TD or OCD are at risk of being affected by other psychiatric comorbidities. For TD, only 10–15% of patients with TD have no comorbidities [6]. The most frequent comorbidities among children and adolescents with TD are attention-deficit/hyperactivity disorder (ADHD) (ranging from 50% to 60%) [3] and OCD (ranging from 11% to 66%) [7,8], and subthreshold obsessive-compulsive behaviors may be up to 6 times more frequent than comorbid OCD [9]. For OCD, up to 80% of children and adolescents with OCD have at least one comorbidity [10], and 21–75% experience two or more comorbid mental disorders [10,11]. Anxiety and depression are the most common comorbid mental conditions of OCD [10,12]. The older patients with OCD are, the higher the rates and severity of comorbid depressive disorders [13,14]. Therefore, the comorbidities of OCD and TD are important dimensions that need to be further investigated.

Furthermore, from a neurobiological perspective, OCD and TS both involve alterations in cortico–striato–thalamo–cortical (CSTC) circuits [15]. In addition to the similarities in symptoms and neurobiology, TS and OCD have correlations in etiology, such as genetics and neurochemistry [16]. Moreover, the co-occurrence of TD and OCD, which appears to be common, poses a particular challenge to clinicians with respect to making treatment recommendations [15].

The epidemiological survey is an important method to investigate the prevalence and comorbidities of diseases. The estimated prevalence of TS ranges from 0.3% to 0.9% in children [17]. A study found that the prevalence rates of OCD in children worldwide are 1–3% [18]. To our knowledge, there are no exact data about TD and OCD in children and adolescents in China. Several local surveys for the TD and OCD of children and adolescents have been conducted. In the Daxing district of Beijing, 4020 students aged 6–16 years were administered TD tests, and the prevalence of TD was 2.26% (TT/chronic motor or vocal tic disorders/TS were 1.05%, 0.73% and 0.47%, respectively) [19]. The prevalence was 13.6% for OCS and 9.04% for OCD by the Leyton Obsessional Inventory—Child Version in a normative Chinese sample of 3221 school students aged 12–18 years old in two Beijing’s administrative districts [20]. The results of Sichuan Province and Hunan Province included in this national survey have been reported as follows: the prevalence of TD and OCD in Sichuan Province with 20,752 students aged 6–16 years old included 2.03% (TT/CMT/CVT/TS were 1.01%, 0.38%, 0.38% and 0.26%, respectively) and 1.09%, respectively [21]; the prevalence of TD and OCD in Central Hunan Province with 17,071 students aged 6–16 years old included 0.62% and 0.57%, respectively [22]. In Hunan Province, China, the most common TD comorbidities are major depressive disorder (MDD) (46.67%), generalized anxiety disorder (GAD) (36.19%), dysthymia (19.05%), ADHD (13.33%), OCD (9.52%), oppositional defiant disorder (ODD) (8.57%), specific phobia (SPP) (6.67%), and conduct disorder (CD) (4.76%) [22]. Regarding comorbidities, a previous study in China reported that over 20% of adolescents with OCD in China had depression, while over 80% had anxiety [14]. In Hunan Province, the most common comorbidities of OCD are ADHD (27.55%), ODD (23.47%), MDD (15.31%), CD (14.29%), TD (10.2%), dysthymia (7.14%), SPP (6.12%) and GAD (1.02%) [22]. There is a lack of enough data to present the prevalence of TD and OCD nationally.

However, the results mentioned above are limited by sample size and location. We might need to determine the prevalence of TD and OCD in children and adolescents. Furthermore, the prevalence of these local surveys may be partly explained using various screening methods, diagnostic criteria, and populations. Last but not least, in the past 40 years, Chinese society has changed tremendously with the implementation of the reform and opening-up policy and economic development, and the mental health of not only adults but also children and adolescents is especially noteworthy [23]. A national survey of the prevalence of mental disorders in children and adolescents in China has been reported in China recently [24]. The results showed that the prevalence of TD and OCD was 2.5% (95% CI: 2.3–3.0) and 1.3% (95% CI: 2.3–3.0), respectively. However, the subtypes of TD and the comorbidities of TD and OCD were not reported.

The purpose of this study was to investigate the prevalence, distribution, and comorbidities of TD and OCD. This was the first nationwide epidemiological survey that aimed to research psychiatric disorders (TD and OCD included) in children and adolescents in China. A two-phase epidemiological survey was conducted, including questionnaire screening and the interview stage. Students aged 6–16 years were enrolled. We used the Diagnostic and Statistical Manual of Mental Disorders, Fourth Edition (DSM-IV) to diagnose mental disorders in children and adolescents.

## 2. Methods

### 2.1. Participants

Chinese school (including primary school and junior and senior middle school) students aged 6–16 years were enrolled in this survey. Older students were excluded due to the College Entrance Examination. According to the level of gross domestic product (GDP), a total of five areas, including Beijing city (BJ), which is representative of a developed urban area, and the other 4 provinces, including Liaoning Province (LN), Jiangsu Province (JS), Sichuan Province (SC), and Hunan Province (HN), are representative of developing areas. Every 2–4 prefecture cities were randomly selected from the five areas, and 15 cities were enrolled. Then, 81 primary schools and 88 middle schools were randomly picked, and in each grade, classes 2–5 were randomly chosen. Finally, a total of 73,992 students in 1764 classes were enrolled in this survey. In Stage 1, we conducted a questionnaire to screen for behavioral problems. The detailed procedure and study protocol are provided in our previous study [24,25].

### 2.2. Screening Scales

The screening instruments used in this study were the Child Behavior Checklist (CBCL) and Mini-International Neuropsychiatric Interview for Children and Adolescents (MINI-Kid). The CBCL and MINI-Kid were mainly used for the first and second stages of the survey, respectively.

CBCL: The CBCL was first used in 1983 by Achenbach and his colleagues and has 118 items rated by parents or caregivers [26]. Each item of behavioral and emotional problem scores ranges from 0 (not true) to 2 (very true). The CBCL has eight factors: withdrawn/depressed, somatic complaints, anxious/depressed, social problems, thought problems, rule-breaking behavior, aggressive behavior, and attention problems [27,28]. The CBCL was administered to the 24,013 city children in 26 units across 22 provinces of China, thus yielding norm scores in 1992 [29]. The Chinese version of the CBCL has been shown to have good reliability and validity [30]. The two-week test–retest reliability and Cronbach’s α of the Chinese version of the CBCL are 0.9 and 0.93, respectively [27]. The parents or other caregivers of the enrolled children or adolescents completed the CBCL [31].

MINI-Kid: The MINI-Kid is a structured instrument for assessing the presence of psychiatric disorders in children and adolescents 6–17 years old and was developed by Sheehan and his colleagues in 1998 [32]. Twenty-four psychiatric disorders (based on DSM-IV and International Classification of Diseases-10) and suicidality were included in the interview. Each disorder is screened by answering 2 to 4 questions. Other symptoms will be asked only if the screen questions are positively answered. All the questions are only answered by “yes” or “no”. The parent version of the MINI-Kid (MINI-Kid-P) was used in this survey. The concordance of MINI-Kid-P and MINI-Kid was good. The MINI-Kid and MINI-Kid–P both have good reliability and validity for assessing psychiatric disorders in children and adolescents. The parent or caregiver and the child both present the interview at the same time. The interrater reliability and test–retest reliability for the Chinese version of the MINI-kid-P were higher than 0.8 and 0.9, respectively [33]. The scale has good reliability and validity and is suitable for epidemiological investigation [34]. The second stage was taken approximately 2 weeks after the first stage. The duration of data collection was from October 2014 to March 2015.

### 2.3. Statistical Analysis

First, the point prevalences of TD (including TTD/CMT/CVT/TS) and OCD were estimated in this study. Second, the sex and age effects on the prevalences of TD and OCD were calculated. Third, we compared the prevalence differences of TD and OCD between sex and age by chi-squared tests. All statistical tests were two-tailed with a significance level of 0.05. SPSS 19.0 was used in this paper. First, sample distributions of sex, age, region, and the screening stages were described. Second, the prevalence of the behavioral and emotional problems in different cohorts was calculated (such as boys vs. girls, young cohort aged 6–11 vs. old cohort aged 12–16). Then, the prevalence of different sexes, regions, and age cohorts was compared with chi-squared tests. For the comparison by age, sex, and region, the analysis of covariance (ANCOVA) and Type III sums of squares were used for the main and interaction effects of age, sex, and region, whereby each significant effect was tested after all other effects were controlled for.

For Stage 1, the product of the sampling weights of respondents’ provincial region, prefectural division, county/district, school, and class were considered. In Stage 2, the weights of randomly picked participants with negative CBCL screening results were multiplied by the reciprocal of their sampling probabilities. Poststratification weight was calculated by age group, sex, and location of residence (urban vs. rural). Nonresponse adjustment was included in the poststratification process. In addition, for missing data, studies indicate that a missing rate of 5% or less is inconsequential [35], and statistical analysis is likely to be biased when more than 10% of data are missing [36]. Therefore, if the missing rate was 5% or less in this survey, the missing data were not included.

### 2.4. The Introduction of the Study Protocol

In this section, we summarize the study protocol of this national survey, which included the following 4 parts. The first part was about how to determine the target sample size; the second part was about the sample procedures; the third part was about the assessment procedure; and the last part was the statistical analysis. For more detailed information, please see the Appendix A (file name: the introduction of the study protocol).

## 3. Results

### 3.1. Sample Description

A total of 73,992 participants aged 6–16 years were enrolled by cluster sampling and screened by CBCL in the first stage. They were from 169 schools (81 primary schools and 88 middle schools; 56 schools from rural areas and 113 schools from urban areas) and 1764 classes. Among them, 36,893 were boys (49.9%), and 49,015 were students from urban areas (66.2%). However, 1885 participants refused to attend the interview. Among them, 925 worried about privacy exposure, 421 thought the survey was unimportant, 277 quit due to too many items of the survey, and 262 submitted incomplete data. In addition, 178 participants were excluded because the CBCL was completed by themselves or their teachers instead of their caregivers. A total of 71,929 completed the CBCL in the first stage. We found missing data with a proportion of 0.35% (less than 0.5%) in the whole sample of 73,992. Therefore, we did not include these missing data.

In the second stage, a total of 14,653 participants were identified as high-risk individuals with mental disorders. In the second stage, a total of 17,524 individuals were included, which was consistent with the 14,653 high-risk individuals and randomly selected groups of 2871 non-high-risk individuals (5.0% of the entire sample) in mental disorders by CBCL for the distribution difference. The mean (standard deviation) age of the whole sample is 11.56 ± 5.12. We list the CBCL scores and their subscale scores in Appendix A. Finally, a total of 13,030 patients were diagnosed with at least one disorder after the MINI-Kid screening and psychiatrists diagnosed according to the DSM-IV. A total of 1772 individuals were diagnosed with TDs, and 986 were diagnosed with OCD (See Figure 1).

### 3.2. Point Prevalence of TD and OCD

The point prevalence of TD was 2.46% (95% CI: 2.35–2.57); the point prevalence of TS was 0.38% (95% CI: 0.34–0.43); the point prevalence of CMTD was 0.45% (95% CI: 0.40–0.49); the point prevalence of CVTD was 0.48% (95% CI: 0.43–0.53); and the point prevalence of TTD was 1.12% (95% CI: 1.08–1.23). The point prevalence of OCD was 1.37% (95% CI: 1.28–1.45) (See Table 1).

### 3.3. Sex Effects on the Prevalence of TD and OCD

Significant differences in the prevalence of TD and OCD existed between sexes (TS: boys: 0.54%, girls: 0.22%; *χ*^2^ = 3.43, *p* < 0.001; TTD: boys: 1.39%, girls: 0.92%; *χ*^2^ = 34.00, *p* < 0.001; OCD: boys: 1.17%, girls: 1.56%; *χ*^2^ = 19.94, *p* < 0.001). No difference was found between boys and girls in the prevalence of TD, CMTD, and CVTD (See Table 2).

### 3.4. Age Effects on the Prevalence of TD and OCD

Statistically significant differences in prevalence of all the TD and OCD groups existed between age groups (TD: children aged 6–12 years old: 3.54%, adolescent aged 13–16 years old: 0.99%; *χ*^2^ = 480.98, *p* < 0.001; TS: children: 0.56%, adolescent: 0.14%; *χ*^2^ = 84.758, *p* < 0.001; CMTD: children: 0.68%, adolescent: 0.14%; *χ*^2^ = 115.44, *p* < 0.001; CVTD: children: 0.72%, adolescent: 0.14%; *χ*^2^ = 125.54, *p* < 0.001; TTD: children: 1.59%, adolescent: 0.58%; *χ*^2^ = 157.65, *p* < 0.001; OCD: children: 1.51%, adolescent: 1.17%; *χ*^2^ = 15.45, *p* < 0.001). (See Table 3). The prevalence rates of TD and OCD showed a sharp downward trend after the age of 12 years old. The highest point prevalence of TD was 9 years old, while the peak prevalence of OCD occurred at 11 years old (see Figure 2 and Figure 3).

### 3.5. Comorbidities of TD, TS, and OCD

Comorbidities of TD, TS and OCD are shown in Figure 3. The prevalence of TD comorbidities was as follows: anxiety disorder (24.94%); ADHD (19.41%); and affective disorder (18.05%). The prevalence of TS comorbidities was as follows: OCD (40.73%); MDD (17.82%); ADHD (14.91%); dysthymic disorder (6.91%); ODD (4.73%); agoraphobia (2.91%); and GAD (2.18%). The prevalence of OCD comorbidities was as follows: TS (11.36%); CVTD (8.4%); dysthymic disorder (4.67%); ADHD (4.26%); MDD (3.35%); CMTD (2.13%); and manic episode (0.41%) (See Figure 4).

In addition, to investigate the age effect and sex effect of OCD, we summarized the number of different types of comorbidities for OCD. For more details, see Appendix A.

## 4. Discussion

This is the first national study on the prevalence of mental disorders in school students aged 6–16 years old in China. The prevalence of TD and OCD was 2.46% (95% CI: 2.35–2.57) and 1.37% (95% CI: 1.28–1.45), respectively. Sex differences existed both in TD and OCD. Specifically, the prevalence of OCD was found to be higher in girls, while the prevalence of TTD and TS was found to be higher in boys, with male-to-female ratios of 1.51:1 and 2.45:1, respectively. A significant age effect also existed in both the TD and OCD groups. The prevalence of all subtypes of TD and OCD was higher among children than among adolescents. The most common comorbidities of TS and OCD were OCD (40.73%) and TS (11.36%), respectively.

Previous studies on the prevalence of TD and its subtypes are in line with this study. A meta-analysis based on school studies found that the prevalences of TTD, CVTD, CMTD and TD were 2.99% (95% CI, 1.60–5.61), 0.69% (95% CI, 0.49–0.97), 1.65% (95% CI, 0.64–4.28) and 0.77% (95% CI, 0.39–1.51), respectively [37]. However, studies have found that transient tics are relatively common, affecting 11% to 20% of school-age children [38,39,40], with a male-to-female ratio ranging from 2:1 to 3.5:1. There are several reasons for this result. The first reason might be the cross-sectional nature of our study, and the use of the MINI-Kid for diagnosis restricted the measurement of currently experienced tic symptoms to the previous month. The second reason might be that the symptoms of TTD are mild and fluctuant, and the initial symptoms may be unnoticed or misdiagnosed [41]. The diagnosis relies on history and observation by the psychiatrist. The third reason for the discrepancies in prevalence may be due to methodological differences rather than true differences [41]. Epidemiological studies of the prevalence of TD range from 0.25% to 5.7% [42]. The Far East has a somewhat lower prevalence rate than other regions, such as the USA and some European countries, perhaps because the Chinese diagnostic system requires both impairment and distress [42].

We found that the prevalences of TS and TTD were higher in boys than in girls, while no difference was found in chronic motor or vocal TDs. This is consistent with previous studies [37]. The prevalence of TD and its subtypes was higher in children than in adolescents. Leckman et al. found that children aged between 7 and 11 years seemed to have the highest estimated prevalence rates of TTD [41]. In this study, we found that the peak ages of the prevalence of TTD were 7 years old, 9–11 years old and 12–13 years old. The peak age of the prevalence of CMTD/CVTD was 11 years, while that of TS was 7–12 years.

Obsessive-compulsive disorder (OCD) is typically regarded as a chronic condition, and relatively little is known about the naturalistic features and longitudinal course of the disorder. This study reported both the prevalence of OCD and its age and sex effect. The prevalence of OCD was 1.367%, which is in line with previous findings obtained from participants of older adolescents [43]. This prevalence is lower than that found in some studies [44,45] and higher than that found in other studies of children and adolescents [46,47]. The current study found that the prevalence of OCD was higher in girls than in boys. This is inconsistent with previous surveys. Some surveys found that no difference existed in the prevalence of OCD between boys and girls [47,48]. Fontenelle et al. found that the prevalence of OCD is higher in female adults and male children and adolescents [49]. In our study, the prevalence of OCD was higher in children than adolescents. We also found that the highest prevalence of OCD occurs at 11 years old, which coincides with the mean age of onset of OCD (11.5 years) [46]. This may be due to the higher prevalence in children (aged from 6 to 12 years) than in adolescents (aged from 13 to 16 years). The prevalence trend of OCD increased with age after 12 years old, while TD decreased rapidly after 12 years old. This coincides with a study that found that OCD initiated at 10–12 years old and often experienced symptom deterioration with age [50]. The tics decreased rapidly in late adolescence and early adulthood [41].

The most common comorbidity of TS was OCD (40.73%), which was consistent with prior studies. Approximately one-third to one-half of children with TS will experience comorbid OCD throughout their lifetime [51]. The study found that 40–60% of patients were diagnosed with TS comorbid with OCD [42]. Comorbid OCD has a worse prognosis than TD alone [52]. Depression is one of the major comorbidities that occurs in TS, with a prevalence ranging from 1.8% to 8.9% [53]. In this study, the most common comorbidity of OCD was TS (11.36%). OCD usually has an onset during the early stage of TD, and it still presents and is more likely to persist than tics in the later stage [3,6].

TS and OCD overlapped in some clinical characteristics. For example, a study found that the cingulate gyrus, temporal gyrus, and medial frontal gyrus were significantly correlated with OCD in TS patients [54]. Premonitory urges were defined as “uncomfortable bodily feelings” and a longing to make things “just right” [55,56]. It precedes tics in TD and compulsions in OCD [55]. Approximately 93% of the patients with TD aged 8–71 years reported premonitory urges [57]. Ferrao et al. found that the majority (65%) of OCD patients experienced premonitory urges rather than obsessions prior to at least one repetitive behavior [58]. The high comorbidity between OCD and TD may be another indication for the physiopathology mechanism and treatment choices.

## 5. Limitations

Several limitations of this study should be acknowledged. The first limitation is the identification of mental health problems only by the CBCL, which is a structured self-rating questionnaire. Second, all the findings were cross-sectional and retrospective, and the symptoms may be influenced by recall bias. Third, the enrolled participants are limited. Students aged 17–18 years old and juveniles who drop out of school should be enrolled in the future to obtain more comprehensive results.

## 6. Conclusions

Our study is the first nationwide survey on the prevalence of TD (2.46%) and OCD (1.37%) in school students aged 6–16 years old in China. The high comorbidity between OCD and TD individuals suggested overlap based on the prevalence dimensions, which might be influenced by age and sex. This result suggested that we also need to pay more attention to the heterogeneity between TS and OCD. TD and OCD often co-occur. This result suggested that we also need to pay more attention to the homogeneity and heterogeneity between TS and OCD. These results suggest that policymakers and mental health service providers should pay more attention to children and adolescents with mental disorders and comorbidities.

## Figures and Tables

**Figure 1 brainsci-12-00650-f001:**
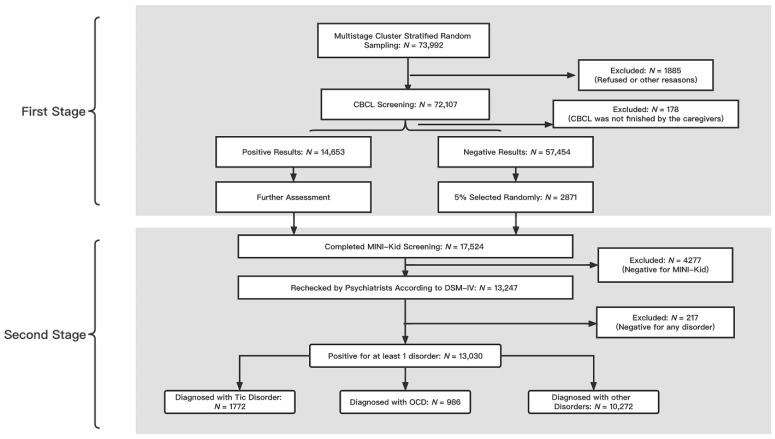
The flowchart for the procedures of this national survey of OCD and TDs (CBCL: Child Behavior Checklist; MINI-Kid: Mini International Neuropsychiatric Interview for Children and Adolescents; OCD: obsessive-compulsive disorder).

**Figure 2 brainsci-12-00650-f002:**
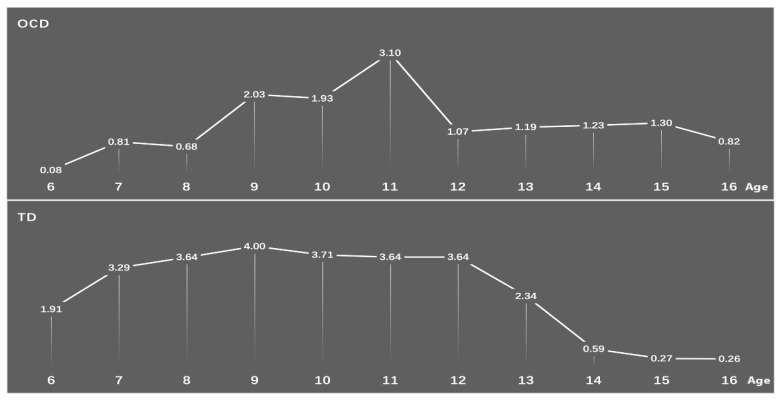
Changes in the prevalence rates of TD and OCD with age (OCD: obsessive-compulsive disorder; TD: tic disorders).

**Figure 3 brainsci-12-00650-f003:**
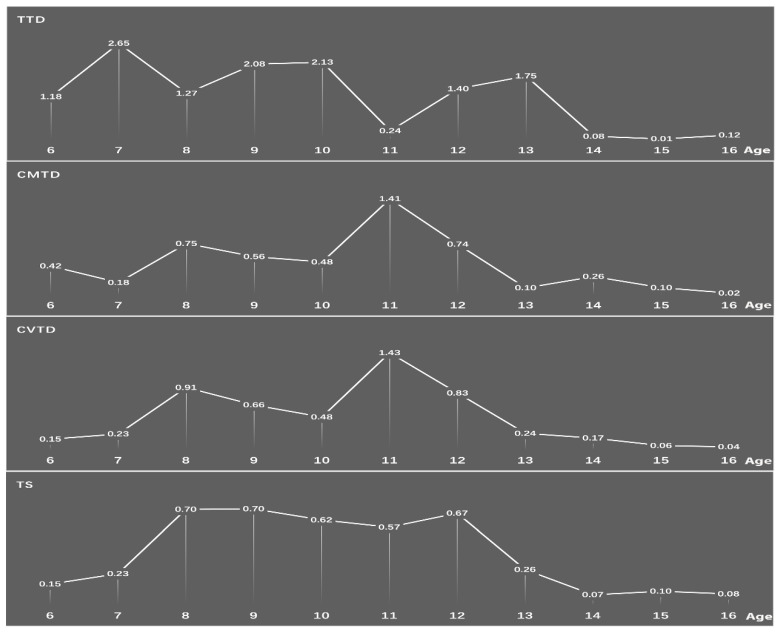
Changes in the prevalence of subtypes of T (TS: Tourette’s syndrome; CMTD: chronic motor tic disorder; CVTD: chronic vocal tic disorder, TTD: transient tic disorder).

**Figure 4 brainsci-12-00650-f004:**
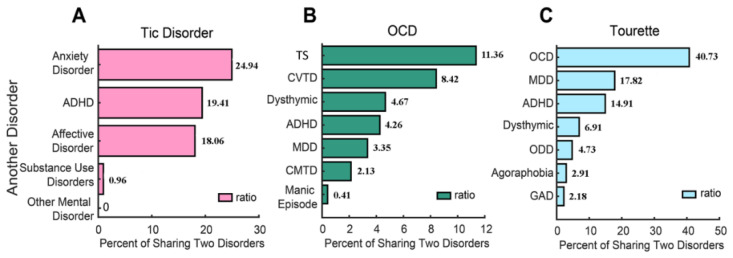
The comorbidities of TD (**A**), TS (**C**), and OCD (%) (**B**) (ADHD: attention-deficit/hyperactivity disorder; OCD: obsessive-compulsive disorder; TS: Tourette’s syndrome; CVTD: chronic vocal tic disorder; MDD: major depressive disorder; CMTD: chronic motor tic disorder; ODD: oppositional defiant disorder; GAD: generalized anxiety disorder).

**Table 1 brainsci-12-00650-t001:** The prevalence of OCD and TD in children aged 6–16.

Mental Disorders	Identified Individuals	Unweighted Point Prevalence (%)	Unweighted 95% CI (%)	Adjusted Point Prevalence (%)	Adjusted 95% CI (%)
OCD	986	1.4	1.3–1.5	1.3	1.2–1.4
TDs	1772	2.5	2.4–2.6	2.5	2.3–3.0
TS	275	0.4	0.3–0.4	0.4	0.3–0.4
CMTD/CVTD	664	0.9	0.8–1.0	0.9	0.8–1.0
TTD	833	1.2	1.1–1.2	1.2	1.1–1.2

Note: OCD: obsessive-compulsive disorder; TDs: tic disorders; TS: Tourette’s syndrome; CMTD: chronic motor tic disorder; CVTD: chronic vocal tic disorder, TTD: transient tic disorder.

**Table 2 brainsci-12-00650-t002:** Sex difference in OCD, TD and TD subtypes.

Disorders	Boys	Girls	*χ* ^2^	*p*
	(*N* = 35,953)	(*N* = 36,154)		
OCD	1.17%	1.56%	19.94	<0.001
Tic Disorder	2.56%	2.35%	3.43	>0.05
TS	0.54%	0.22%	47.27	<0.001
CMTD	0.45%	0.45%	<0.001	>0.05
CVTD	0.51%	0.44%	1.68	>0.05
TTD	1.39%	0.92%	34.00	<0.001

Note: OCD: obsessive-compulsive disorder; TS: Tourette’s syndrome; CMTD: chronic motor tic disorder; CVTD: chronic vocal tic disorder, TTD: transient tic disorder.

**Table 3 brainsci-12-00650-t003:** Age difference in OCD, TD and TD subtypes.

Disorders	Children	Adolescents	*χ* ^2^	*p*
	(*N* = 41,332)	(*N* = 30,775)		
OCD	1.51%	1.17%	15.45	<0.001
Tic Disorder	3.55%	0.99%	480.98	<0.001
TS	0.56%	0.14%	84.76	<0.001
CMTD	0.68%	0.14%	115.44	<0.001
CVTD	0.72%	0.14%	125.54	<0.001
TTD	1.59%	0.58%	157.65	<0.001

Note: OCD: obsessive-compulsive disorder; TS: Tourette’s syndrome; CMTD: chronic motor tic disorder; CVTD: chronic vocal tic disorder, TTD: transient tic disorder.

## Data Availability

The data presented in this study are available on request from the corresponding author. The data are not publicly available due to privacy.

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
