# Peer review of "The Prevalence and Comorbidity of Tic Disorders and Obsessive-Compulsive Disorder in Chinese School Students Aged 6–16: A National Survey"

_brainsci, 2022, doi:10.3390/brainsci12050650_

Round 1

Reviewer 1 Report

Many comments have been responded and the manuscript has been improved, however here some revision is needed again:

Abstract:

The statement of Background makes this manuscript local as if it is not appropriate for a non-Chinese journal.

The conclusion must include something other than the results

Introduction:

The fact that obsessive-compulsive disorder or Tick has not been investigated in China for 40 years ago, is not a sufficient reason for the rationale of the current study. The main question has remained without response: why have you studied Tic disorder and OCD with each other?  Do both of these disorders have the same evolutionary-brain processes? Do they have the same etiology? and so on...

Methods

Please provide a sampling diagram.

The authors have said that they have deleted missing data, why? Has the deletion of this data led to a bias in the results? explain by reasoning and using the statistical analysis method of missing data.

Results:

In Tables 1 and 2, please report the frequency of each percentage and also the confidence interval for all.

Please report frequency and percentage statistics both by weighting and raw data.

conclusion

The conclusion must include something other than the results.

Reviewer 2 Report

The article was partially revised. But in the method section, the tools have not been fully and separately explained yet. Please about the tools, who made it, what year it was made, scoring, reliability, and validity.

Please do not repeat the findings. Rewrite the conclusion accordingly: Summarize the research, mention the benefits of future research, provide solutions, and point to larger research.

Author Response

This manuscript is a resubmission of an earlier submission. The following is a list of the peer review reports and author responses from that submission.

Round 1

Reviewer 1 Report

The study is much needed; however, the description of the methods is severely lacking in detail as is the description of the sample. These issues need to be resolved. My comments are listed below.

Introduction

Please add a reference after “TD and OCD often cooccur and overlap to some extent.”

Line 42: What exactly is meant by “borderline symptoms”. Consider rephrasing. 

Line 47: “This indicates that the comorbid condition of TD may affect the prognosis more than the disease itself.” Unclear what is meant by this sentence. Prognosis of TD or OCD? Please clarify.

Line 67: “The estimated prevalence of TS ranges from 3 to 8 per 1,000 in school-age children [1].” Please refer to original studies rather than the DSM manual. 

All abbreviations, including the ones for psychiatric disorders, need to be written in full for the first time they are used. 

The section about TD and OCD prevalence could benefit from being restructured/rewritten to give the reader a better understanding of the quality and size of the studies mentioned. Also, you mention several surveys and then state that this is the first survey to evaluate the prevalence. It needs to be made clearer what the current study does that was not done with the previous surveys in order to justify this statement. 

Method
Line 113: “In the second stage, the MINI-Kid-based and DSM-IV-based interviews were successively conducted by two different trained psychiatrists to recognize the diagnoses, and both positive results were needed for the final diagnoses” It is unclear how exactly this was done and what the DSM-IV interview consists of. Were the interviews performed on all participants? If not, how many were interviewed? 

The description of the procedures needs to be much more detailed and separated from the scale descriptions. All stages of the study need to be described clearly. 

Results
The description of the sample inclusion stages belongs in the method section. 

Please provide a more detailed description of the sample including age, gender etc. 

The manuscript would benefit from proof reading by a native English speaker. 

Please write all statistical symbols in italic. 

Reviewer 2 Report

One of the strengths of this study is the large sample size, Although, there are many corrections needed:

Introduction:

All content needs to be more organized.

 The authors should rationalize why this study is significant.

Methodology:

The procedure of the study is not clear. The protocol of the study should be presented briefly.

How did you calculate the sample size?

 It seems the geographical diversity in China so the data analysis should be performed by weighting.

Psychometric properties of the measures in China and for the study population should be mentioned.

The study seems to have screening and diagnostic phases, how to access the sample in each stage.

Introduce the variables in the study.

This study has very important sources of biases that are not mentioned.

How did you handle missing data?

Results:

It is necessary to present all comorbid disorders with obsessive-compulsive disorder by gender and age group.

The age trend of both disorders should be presented separately in a graph.

Comparisons based on two groups of children and adolescents do not seem to provide much information

The current findings have not prepared the role of covariate variables in the prevalence of each disorder.

Discussion

More scientific explanations are needed as to why the trend differences between the two disorders in the two groups.

Reviewer 3 Report

Report the method of data analysis in the research method section in the abstract.

In the introduction, explain the characteristics of the research problem and the research gap.

In the research method section, the tools should be fully described and the reliability of the tools in this research should be reported.

Conclusion The end of the article should be changed and not the repetition of the content of the results but should be written based on the results of the article.